# Is Users' Trust during Automated Driving Different When Using an Ambient Light HMI, Compared to an Auditory HMI?

**Rafael Cirino Gonçalves [1,*], Tyron Louw [1], Yee Mun Lee [1], Ruth Madigan [1], Jonny Kuo [2], Mike Lenné [2] and Natasha Merat [1]**

1   Institute for Transport Studies, Faculty of Environment, University of Leeds, Leeds LS2 9JT, UK
2   Seeing Machines, Melbourne, VIC 3066, Australia
*   Correspondence: trarg@leeds.ac.uk; Tel.: +44-791-928-2338

**Abstract:** The aim of this study was to compare the success of two different Human Machine Interfaces (HMIs) in attracting drivers' attention when they were engaged in a Non-Driving-Related Task (NDRT) during SAE Level 3 driving. We also assessed the value of each on drivers' perceived safety and trust. A driving simulator experiment was used to investigate drivers' response to a non-safety-critical transition of control and five cut-in events (one hard; deceleration of 2.4 m/s², and 4 subtle; deceleration of ~1.16 m/s²) over the course of the automated drive. The experiment used two types of HMI to trigger a takeover request (TOR): one Light-band display that flashed whenever the drivers needed to takeover control; and one auditory warning. Results showed that drivers' levels of trust in automation were similar for both HMI conditions, in all scenarios, except during a hard cut-in event. Regarding the HMI's capabilities to support a takeover process, the study found no differences in drivers' takeover performance or overall gaze distribution. However, with the Light-band HMI, drivers were more likely to focus their attention to the road centre first after a takeover request. Although a high proportion of glances towards the dashboard of the vehicle was seen for both HMIs during the takeover process, the value of these ambient lighting signals for conveying automation status and takeover messages may be useful to help drivers direct their visual attention to the most suitable area after a takeover, such as the forward roadway.

**Keywords:** automated vehicles; transition of control; trust in automation; human machine interfaces; gaze behaviour; Light-band displays

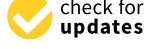



## 1. Introduction

The latest automated driving technology currently available on the market assumes both lateral and longitudinal vehicle control, allowing the driver to physically disengage from the driving task (SAE L2, SAE, [1]). However, these automated driving systems still require the driver to monitor the driving task and surrounding environment and intervene when necessary or when requested to do so by the system [1]. The technology enabling higher levels of automated driving is constantly developing, with some flagship vehicle models now equipped with SAE Level 3 automated driving systems that are legal to be used in some road and driving conditions, e.g., the newly-released Mercedes-Benz Drive Pilot [2]. These systems (and current recommendations—e.g., SAE, 2021) allow the driver to physically and cognitively disengage from the driving task while automation is active, engaging in Non-Driving-Related Tasks (NDRTs), such as reading an email or text message. However, drivers are still responsible for resuming control if requested by the system, for example, due to an operational limitation (e.g., road works, inclement weather), the end of the trip, or a system malfunction.

However, one of the primary human factors challenges for Level 3 (L3) automated driving systems (as defined by [1]) is ensuring that drivers are capable of responding to requests to intervene (RtI) in a timely and safe fashion. The main touted benefit of L3

automated driving is that the driver can engage in other activities (NDRTs), which is likely to take their eyes (and mind) away from the forward roadway and the main driving task. Simulator and test-track studies with L2 driving have shown that, if drivers spend too much time looking away from the forward road, they are more at risk of colliding into an impending lead obstacle, especially if the driver is not expecting to resume control from the vehicle or not given sufficient time to do so [3]. Level 3 functionalities, on the other hand, are expected to provide drivers with sufficient time to take over control, bringing them back into the control loop, with a suitably long time budget for a response (currently recommended to be around 10 s [4]). In this case, drivers should feel comfortable to engage in NDRTs while the automation is engaged. This may pose a challenge for system designers, ensuring that the automated driving system engenders the appropriate level of trust that balances the driver's desire to engage in an NDRT against the system's limitations and its requirement that the driver responds to a takeover or request to intervene (RtI). One approach to tackle this issue is to somehow keep drivers in the "control loop" by informing them of the automated system's status, allowing them to engage in NDRTs while the automation is engaged, helping them to maintain enough situation awareness [5] to safely resume control, if needed. In this paper, we compare driver response to a takeover request (TOR) by a more traditionally used auditory alert with a novel peripheral light-based display, intended to inform drivers of automation status, enhancing their trust in the system during automated driving and facilitating a smooth transition between L3 and manual driving.

In today's vehicles, system status and functionality, including engagement of driver assistance systems, is primarily communicated through an instrument cluster or Human-Machine Interface (HMI), normally placed behind the steering wheel or on a central console. This information is within the drivers' line of sight and allows them to detect potential incoming hazards with their peripheral vision [6]. Using a manual driving simulator study, ref. [7] found that engaging in a self-paced, visual–manual tracking task with high visual eccentricity (60 degrees) still allowed drivers sufficient on-road glances, suggesting that drivers use peripheral vision to collect evidence for braking during off-road glances.

However, the design solution to convey information about an L3 automated system behind the steering wheel may present issues, given that drivers are unlikely to be facing the road, but rather would be facing the interior of the vehicle, engaging with an NDRT. When drivers are engaged in a visual NDRT, it is difficult for them to detect any changes in the visual information provided by the dash-based HMI. Therefore, they often need to shift their visual attention between the dash area and the NDRT, to confirm system status or an RtI. This is at odds with the intended user-related benefits of higher levels of automation (i.e., comfort and ability to engage in NDRTs). Moreover, if drivers are engaged in an NDRT, with their eyes away from the forward roadway and the dash area, they may need to be alerted with auditory cues to attend to more critical situations, for example, if the system reaches a technical limitation.

Relying on auditory cues alone is potentially problematic, since these could be masked by other auditory noise in the environment, or their meaning could be difficult to decipher amongst a range of similar auditory alerts. Therefore, to enhance comprehension, system notifications/alerts are typically multi-modal, e.g., audio-visual. However, as discussed above, as drivers in L3 automation are allowed to be engaged in NDRTs, they alter their posture and direct their visual and cognitive attention away from the main driving task and road environment. This also means that they may miss the visual element of an audio-visual alert, which is normally placed in an area that would be perceived by a forward-facing driver. This could be problematic for Automated Vehicles (AVs) if system limitations cannot detect sudden or unexpected obstacles which may have been noticed by a forward facing, alert driver. In a manual driving study, ref. [8] showed that mean time-to-collision to a decelerating lead vehicle decreased significantly with increasing eccentricity of an in-car foveal task when measuring brake response for each position of the in-car task. Therefore,

the placement and saliency of existing visual alerts may not be appropriate or perceptible in this new context, where drivers may be out of the traditional driving position in L3 AVs.

According to [9], two of the core design principles of human–automation interaction are to keep drivers in the loop where possible and to support smooth re-entry into the loop. If a system does not allow drivers to maintain adequate system updates or detect important system notifications, it would fail to adhere to these principles and may cause drivers to interrupt their NDRT to check system status, which may lead to lower acceptance and, therefore, use of the system. Studies from [10] have shown that inappropriate levels of trust may change the way humans interact with an automated system, enforcing them to perform more frequent system information checks and therefore increasing the levels of workload for the operator.

In recent years, a number of novel HMI concepts have been proposed to both keep the driver in the loop and support drivers' re-entry into the loop after automated driving. These include tactile [11], visual [12], auditory (including speech- and non-speech-based [13]), and even olfactory cues [14]. Given drivers' likely high engagement in visual NDRTs in the future, the use of ambient Light-Emitting Displays (LEDs) has been explored as a means to provide peripheral light-based cues to drivers about an AV's current state, along with requests for a driver's attention or action. These solutions come after evidence from allied domains suggest that the peripheral visual channel can be used to relay system-related information when the operator is not monitoring the system. In the aviation domain, for example, ref. [15] assessed the effectiveness of current foveal feedback and two implementations of peripheral visual feedback for keeping pilots informed about changes in the status of an automated cockpit system. They found that peripheral visual displays resulted in higher detection rates and faster response times without interfering with the performance of concurrent visual tasks any more than currently available automation feedback.

In the driving context, peripheral ambient light displays have been investigated as potential collision warning tools [16], as lane change decision aids [17], as a means to help modulate drivers' speed [18,19], to guide drivers' attention for identifying targets (road users/obstacles), and for indicating vehicle intention [20,21]. Peripheral ambient light displays have also been used to inform drivers of malfunctioning Advanced Driver Assistance Systems (ADAS) [22] and to facilitate collaborative driving tasks between the driver and the co-driver [18].

Recently, light-based displays have also been applied in the context of automated driving. For example, ref. [23] conveyed contextual information through ambient displays, to assist drivers during takeover requests, and found that this resulted in shorter reaction times and longer times to collision, without increasing driver workload. More commonly, light displays have been able to provide information/warnings to drivers about other road users or the AV's intentions [24]. Research from both manual and automated driving shows that, in general, ambient lights are rated highly by drivers and drivers are responsive to these peripheral cues [17]. Therefore, adding a peripheral light display indicating the current automation state may have potential benefits in terms of reducing the frequency of drivers' glances to the road during automation, increased engagement in NDRTs, and higher perceptions of trust in and comfort during automated driving. However, few studies have investigated the use of these displays to improve drivers' perceptions of trust and safety during automated driving and to facilitate transitions between L3 automation and manual driving.

Therefore, the current driving simulator study addressed this gap via the following research questions.

1. Can an ambient peripheral light display (Light-band HMI) be used to improve drivers' perceived safety and trust during L3 automated driving?
2. How effective is a Light-band HMI for facilitating an effective transition of control between L3 automated driving and manual driving, when compared to an auditory HMI alert?

3.  What is the pattern of drivers' eye movements during the takeover process for each type of HMI?

## 2. Methods

### 2.1. Participants

Following approval from the University of Leeds Research Ethics Committee (Reference Number: LTTRAN-132), we recruited 41 drivers via an online social media platform. Participant demographic details are displayed in Table 1. Participants received GBP 30 for taking part in the experiment and were free to withdraw at any point.

**Table 1.** Participant demographics information.

|  | Males (N = 20; Mean (SD)) | Females (N = 21; Mean (SD)) |
|---|---|---|
| Age (years) | 44 (13) | 44 (13) |
| Years with license | 25 (13) | 24 (12) |
| Miles driven annually | 10,300 (5332) | 6642 (3350) |

### 2.2. Equipment

The experiment was conducted in the fully motion-based University of Leeds Driving Simulator (UoLDS), which consists of a Jaguar S-type cab housed in a 4 m-diameter spherical projection dome with a 300°-field-of-view projection system. The simulator also incorporates an 8-degrees-of-freedom electrical motion system. This consists of a 500 mm-stroke-length hexapod motion platform carrying the 2.5T payload of the dome and vehicle cab combination, allowing movement in all six orthogonal degrees of freedom of the Cartesian inertial frame. Additionally, the platform is mounted on a railed gantry that allows a further 5 m of effective travel in surge and sway. Inside the simulator's vehicle cabin, a Lilliput 7″ VGA touchscreen with 800 × 480 resolution was installed near the gear shift and used for a non-driving-related, secondary task, described below.

A Seeing Machines driver-monitoring system was used to record the participants' eye movements, at 60 Hz. Figure 1 shows the system's world model from the driver's point of view, with annotated areas of interest. Included in this world map is the On Road area, which encompassed the full width of the road in the drivers' current and projected future path.

### 2.3. Experimental Design

In this experiment, participants completed two experimental drives on a three-lane UK motorway with ambient traffic. Each experimental drive lasted ~17 min, with five ~2 min automation segments interspersed with ~1 min manual driving segments (Figure 2). There were five non-critical takeover requests per drive and 10 takeovers in total. The entire experiment lasted approximately 2 h. In both experimental drives, during automated driving, participants were instructed to engage in a visual non-driving-related "Arrows" task (NDRT [25]). The Arrows task requires participants to search for, and touch, the upward-facing Arrow, displayed in a 4 × 4 grid of Arrows, using a touch screen in the centre console. Each time the upward-facing arrow is correctly identified and selected, a new grid of arrows is generated. The screen displayed the current participant's cumulative score and a 'score to beat' to keep them engaged in the task. Participants were informed that they would receive an additional GBP 10 if they managed to beat this score, in addition to the agreed GBP 20 received for their participation. However, due to ethical reasons, all participants received the full payment of GBP 30 at the end of the experiment, regardless of their final score on the Arrows task. Analysis of performance in the Arrows task indicated drivers had constant attention to the NDRT throughout the automation period.

**A**

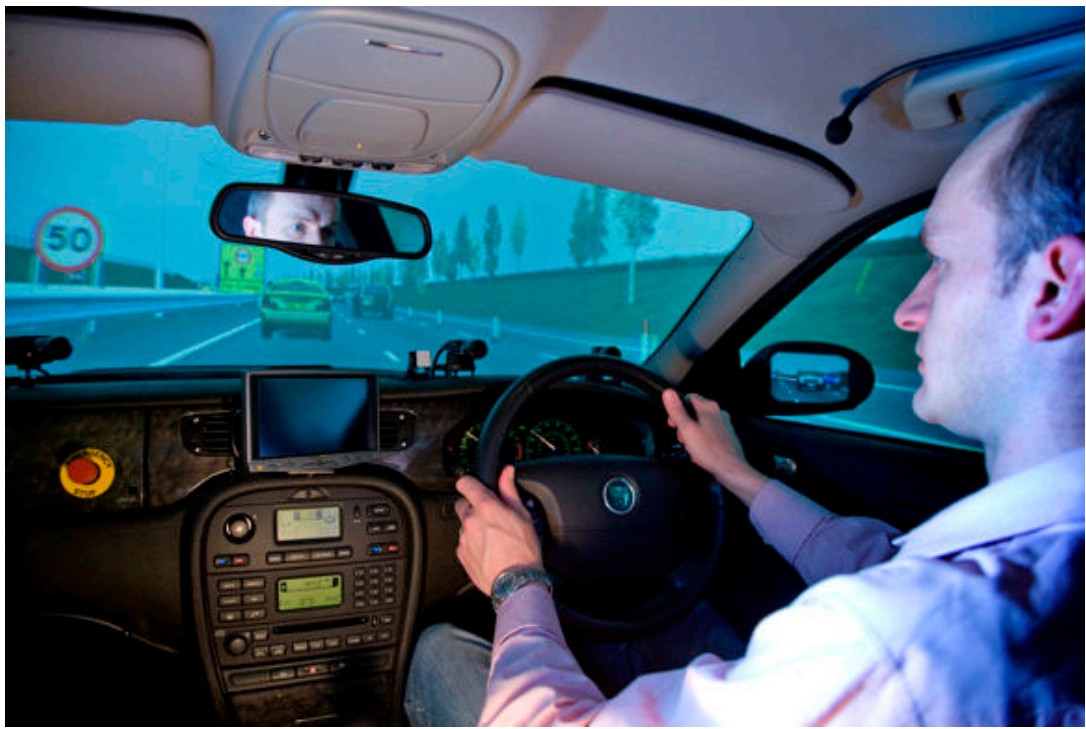

**B**

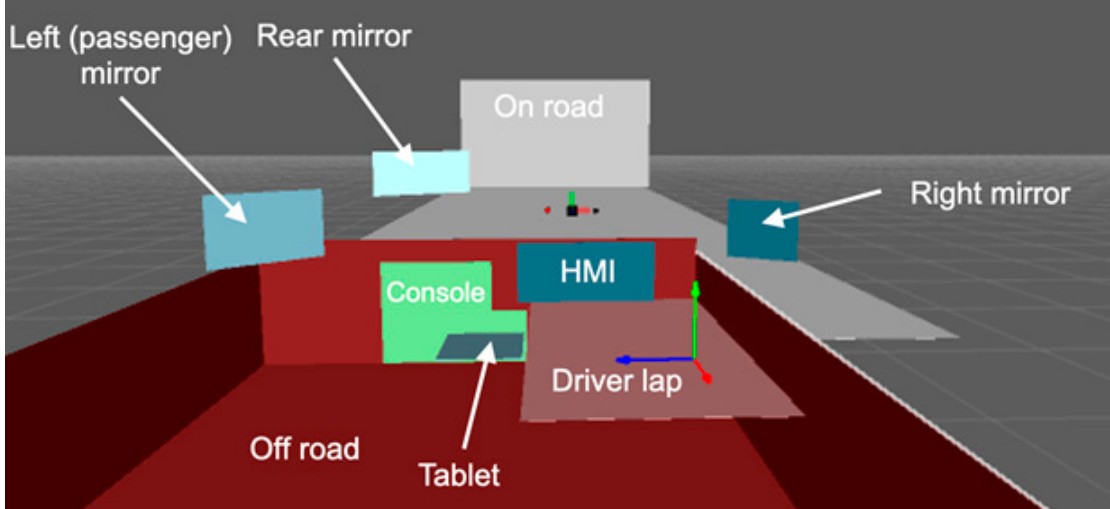

**Figure 1.** Visual representation of the interior of the simulator cab (**A**), and a screenshot of the Seeing Machines driver-monitoring system world model, with annotated areas of interest (**B**).

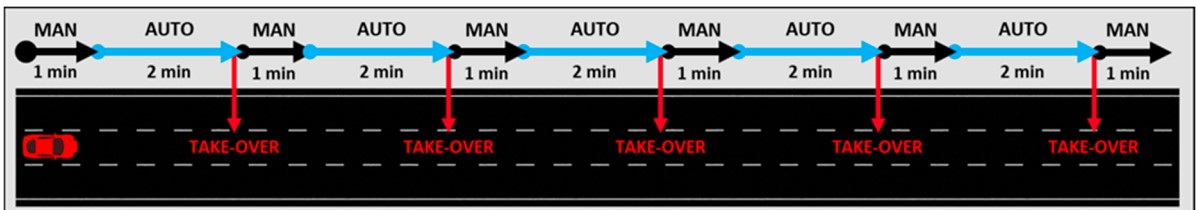

**Figure 2.** Schematic representation of each experimental drive, with five 2 min automation driving segments interspersed with six 1 min manual driving events, separated by five takeover events.

A 2 × 5 within-participant design was used for this study, with the factors HMI type (Light-band, Auditory) and takeover number (1–5), which specified the number of times drivers resumed control during the experimental drive, for each HMI condition (Figure 2).

HMI type was fully counterbalanced across participants and specifies the HMI that the drivers were presented with during automated driving and used for the takeover, i.e., Light-band HMI or Auditory HMI. Selection of these two HMIs was motivated by previous work [3] suggesting that drivers in L3 automation do not continue to monitor the environment and the vehicle's dashboard (where information about system status is normally located) during an automated drive. Therefore, efficient RtI strategies must be able to draw drivers' attention away from other activities (e.g., NDRTs) using salient stimuli, reducing the need for drivers to constantly scan the dash-based interface. For this reason, once drivers' attention was captured by one of the TORs (Auditory or Light-band), we provided the same dash-based information for both experimental conditions after the RtI in order to keep the additional information provided by the dash-based HMI constant (see Figure 3). Our aim was to compare the benefit of the LED and Auditory stimuli for grabbing drivers' attention, keeping any additional information from the dashboard after the TOR constant.

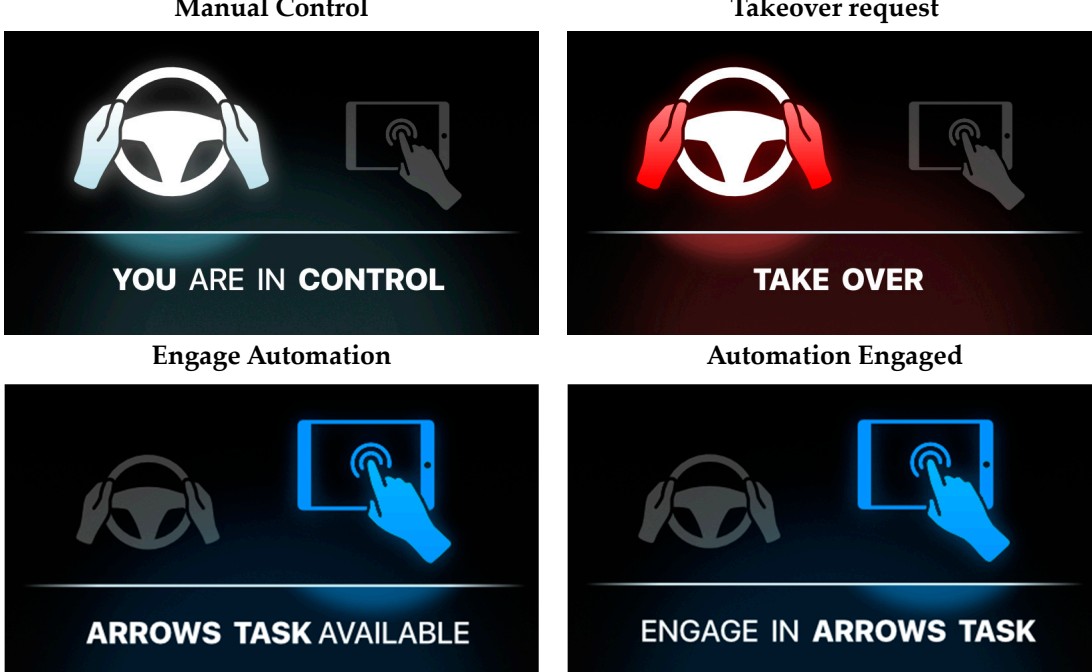

**Figure 3.** The dash-based HMIs used for the study.

In terms of the instructions provided to drivers for takeover, the same text and symbols (Figure 3) were displayed in the vehicle's dashboard display for both conditions. The dashboard symbols for "Takeover request" and "Engage automation" pulsed at a rate of 2 Hz (in parallel with either the Light-band HMI or Auditory HMI) until the driver resumed control or engaged automation, as required. When active, the display of the symbols for "Manual control" and "Automation engaged" remained constant.

For the Light-band condition, an LED-based Light-band notification system was displayed in the vehicle cabin during automated driving and for signalling takeovers. As shown in Figure 4, the Light-band was placed horizontally along the top of the whole dashboard. To allow a clear distinction between the different stages of a takeover, three different settings were used for the Light-band, which were briefly piloted before the study:

1. When automation was available to be engaged, the Light-band pulsed with a blue light at 2 Hz until the driver turned the automation on.
2. During automated driving, the Light-band displayed a solid blue light to indicate that the automation was operating normally.
3. During takeover requests, the Light-band pulsed with a red light at 2 Hz until the driver resumed manual control.

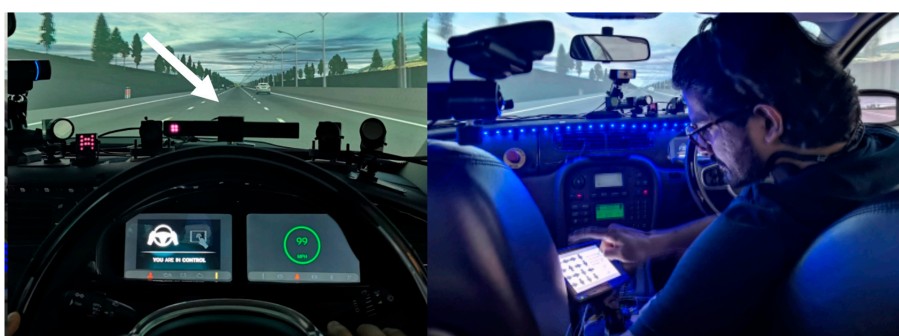

**Figure 4.** Placement of the automation status symbol and vehicle speed in the dashboard display (**left**), and the driver performing the Arrows task during automated driving in the Light-band HMI condition (**right**). The left picture also shows the Seeing Machines driver-monitoring system on the dashboard, above the steering wheel (white arrow).

The Light-band HMI was not accompanied by any auditory warnings. During manual driving, the Light-band was not active. For the Auditory HMI condition, participants only received an auditory alert (880 Hz, lasting 0.2 s) for engaging/disengaging the automated driving system. No other stimuli were provided during automation in this condition.

### 2.3.1. Vehicle Cut-In Scenarios

To assess if drivers' desire to engage in the NDRT would be disturbed by changes in the driving environment, and whether this engagement/disturbance would be different for the two HMI conditions, we implemented a number of vehicle cut-in scenarios when automation was engaged. Approximately 60 s into each 2 min automated drive, a vehicle entered the driver's lane from the left-hand lane. This lane change occurred when the time-to-collision between the subject car and the lead vehicle was either 10 s (events 1, 2, 4, and 5) or 4 s (event 3). For events 1, 2, 4, and 5, the desired headway of the subject car was 1.5 s, resulting in a deceleration of 1.14–1.19 $m/s^2$ during the cut-in scenario, which was a subtle, but sufficient, cue to indicate the presence of the lead vehicle. However, for event 3, the headway was 0.5 s, resulting in a deceleration of 2.4 $m/s^2$, which was a much more salient cue. We hypothesised that each cut-in would signify a change in the driving environment and act as a cue for drivers to pay more attention to the road ahead, perhaps indicating the need to resume control. We assumed that, if drivers trusted the automated system, they would look up to the scene less often during these cut-in scenarios, apart from the more salient event 3. We also hypothesized that, because Light-band provided information about automation status, it would encourage continued engagement in the NDRT, reducing the need for drivers to look up and ensure the automation was still active.

### 2.4. Driver Disengagement Algorithm

At higher levels of automation, drivers are allowed to engage in NDRTs, but they may still be required to resume control from the vehicle, e.g., at the end of an ODD. In these situations, it is important to ensure that the human driver is fully ready to reengage in the driving task and provides an appropriate level of attention and control, with their hands on the wheel and their eyes and attention to the forward roadway. Therefore, in this study, we developed an algorithm that determined when it was safe to hand control back to the driver. This also prevented any accidental/unwanted disengagements. The algorithm used information from the touch-sensitive steering wheel and eye-gaze data from the driver monitoring system to determine in real time whether the driver had their hands on the wheel and was looking at the road ahead. When the takeover request was issued, the system would only disengage if the participant:

1.  Was pulling the stalk to turn automation off.
2.  Had at least one hand on the steering wheel (determined via the capacitive steering wheel).

3.  Was looking ahead (i.e., On Road) for a sufficient length of time (see below for thresholds).

All three criteria had to be met before disengagement was possible. We implemented two different thresholds for glances ahead (criterion 3 above) depending on whether the participant was already looking On Road.

Threshold 1: if, at the point criteria 1 and 2 were met, the participant was not looking On Road, then at least 60% of the glances in the previous 2 s would need to be On Road before the algorithm would consider that criterion 3 was met.

Threshold 2: if, at the point criteria 1 and 2 were met, the participant was already looking On Road, then at least 40% of the glances in the subsequent 2 s would need to be On Road before the algorithm would consider that criterion 3 was met.

*2.5. Procedure*

As part of the recruitment process, participants were emailed a screening and demographics questionnaire, which included questions about age, gender, driving experience, and their experience with different Advanced Driver Assistance Systems (ADAS). To be eligible for taking part in the experiment, participants had to hold a valid driving license, have at least one year's experience driving in the UK, and not have participated in a driving simulator study that included interaction with automated vehicles. Prior to arrival, participants were emailed a description of the study and information about COVID-19 procedures during the experiment and were asked to sign a consent form.

Upon arrival at the simulator, the experimenter asked the participant a series of questions to ensure COVID-19 compliance. They were then taken into the building, where the experiment was explained in more detail, and they were given the opportunity to ask questions. Participants were then taken into the simulator dome and the experimenter explained all the safety procedures, the driving controls of the vehicle, and various dashboard icons and gave a description of how to do the Arrows task, as well as engage and disengage the automated driving system. To enable automation, participants were asked to drive in the centre of the middle lane, maintaining the 70 mph speed limit and adhering to the standard rules of the road, ensuring safe operation of the vehicle, throughout the drive. When active, the automated driving system (ADS) assumed lateral and longitudinal vehicle control and maintained a maximum velocity of 70 mph.

Before each of the two experimental drives, participants performed a short practice drive. To avoid confusing participants by showing them both takeover HMIs at the start of the experiment, they were only shown the HMI system that they would experience in the subsequent experimental drive (i.e., Light-band or Auditory HMI).

They were then left in the simulator dome to perform the first practice drive, allowing them to become familiar with the simulator controls and motion system. The experimenter talked with the participant via an intercom system, when required.

The experiment began with the participant driving in manual mode for a couple of minutes, after which they received an instruction from the automated driving system to turn the automation on. This was achieved by pulling the left indicator stalk towards them. Once automation was engaged, participants began performing the Arrows task. After approximately 2 min from the start of each automated driving segment, participants received a notification to take over control. Participants had to meet the criteria of the disengagement algorithm to turn the automation off. To allow a non-critical takeover, there was no lead vehicle or obstacle during the takeovers.

After the practice drive, and after each experimental drive, participants took a short break while they were seated in the driving simulator. At this point, they were asked to complete a short set of questions, outlined in Table 2. The questionnaire assessed drivers' trust and acceptance of the driving automation system for each of the two takeover HMIs. This questionnaire was tailored specifically for this experiment as part of a series of studies for the L3Pilot project [26], and was validated by a group of experts from the project consortium.

**Table 2.** Experiment questionnaire.

| Question | Response Format | Baseline (Pre-Experiment) | Post Drive 1 | Post Drive 2 |
|---|---|---|---|---|
| I trust that the vehicle will drive safely, while I do the Arrows task | 5-point scale (Strongly disagree-Strongly agree) | x | | |
| I trusted that the vehicle would drive safely while I did the Arrows task | 5-point scale (Strongly disagree-Strongly agree) | | x | x |
| If your level of trust in the automated driving system changed since the start of the experiment, please explain why. | Free text | | x | x |
| I felt safe while doing the Arrows task during automated driving | 5-point scale (Strongly disagree-Strongly agree) | | x | x |
| In this drive, the Light-band/Auditory signal was... | 5-point scale for each Van der Laan Scale item | | x | x |
| How engaged were you with the Arrows task while automation was on? | 10-point (Not at all engaged-Highly engaged) | | | x |
| Apart from takeover requests, was there anything that interrupted your engagement in the Arrows task while automation was on? If so, please explain briefly. | Free text | | | x |
| Which warning system did you prefer? | Light-band/Auditory | | | x |

*2.6. Statistical Analyses*

We analysed data with SPSS V.24 (IBM, Armonk, NY, USA), and generated the visualizations in R and Microsoft Excel. An α-value of 0.05 was used as the criterion for statistical significance, and partial eta-squared was computed as an effect size statistic. Unless otherwise stated, variance of the data was homogenous, as assessed by Levene's test of equality of error variance.

## 3. Results

*3.1. Visual Attention during Automated Driving*

We were interested in understanding whether the Light-band HMI increased participants' understanding of automation status and encouraged their trust in its operation, resulting in a more sustained engagement in the Arrows task during the lead vehicle cut-in events, when compared to the Auditory HMI drives (note that the Auditory HMI was only used during takeover requests). To assess this, we calculated drivers' visual attention to the On Road area after each cut-in event and conducted a Chi-Squared analysis on the number of drivers who made at least one fixation to the On Road area of interest (see Section 2.3.1) in the 10 s after the lead vehicle braked, comparing between the two takeover HMIs, across the five lead vehicle cut-in events. Fixations were calculated using a 200 ms threshold, with a standard deviation below 1° for the gaze position [27].

Results showed no difference in the number of fixations to the On Road area [$X^2$ = 2.35, $p$ = 0.125] between the two HMI conditions. For both conditions, the proportion of drivers who looked On Road during the cut-in events was relatively low (11% of drivers made at least 1 fixation). There was a significant difference in fixations between the five lead vehicle cut-in events [$X^2$ =37.226, $p$ =0.0001], with many more fixations to the On Road area after the third event (Figure 5).

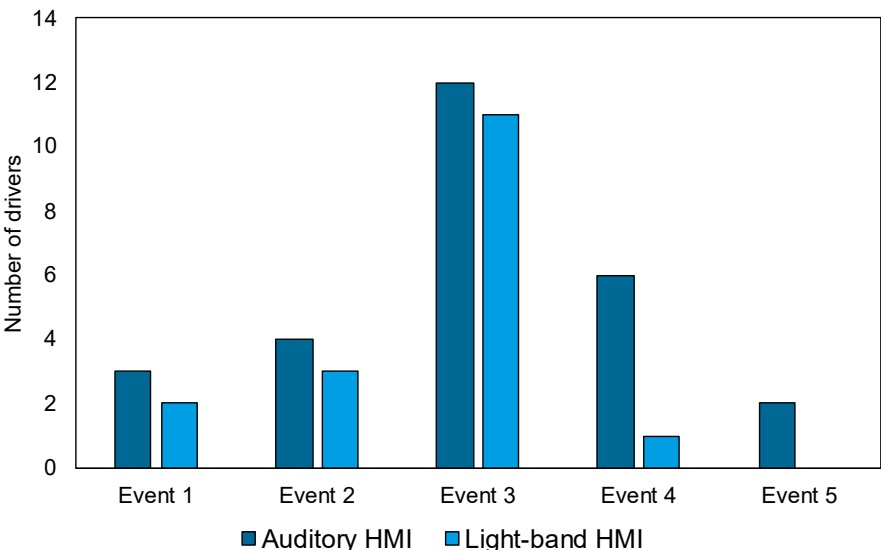

**Figure 5.** Number of drivers who fixated on the road at least once in the 10 s after a cut-in event.

*3.2. Behaviour during the Takeover*

To assess the ability of either takeover HMI to capture drivers' visual attention during the takeover, we compared the distribution of drivers' visual attention allocation to different areas of interest (AoIs) (see Figure 6), as well as how long it took drivers to make a fixation to the On Road area and Instrument Cluster, after the TOR. We also examined how long it took drivers to place their hands on the steering wheel (hands on wheel time) and disengage automation (automation disengagement time).

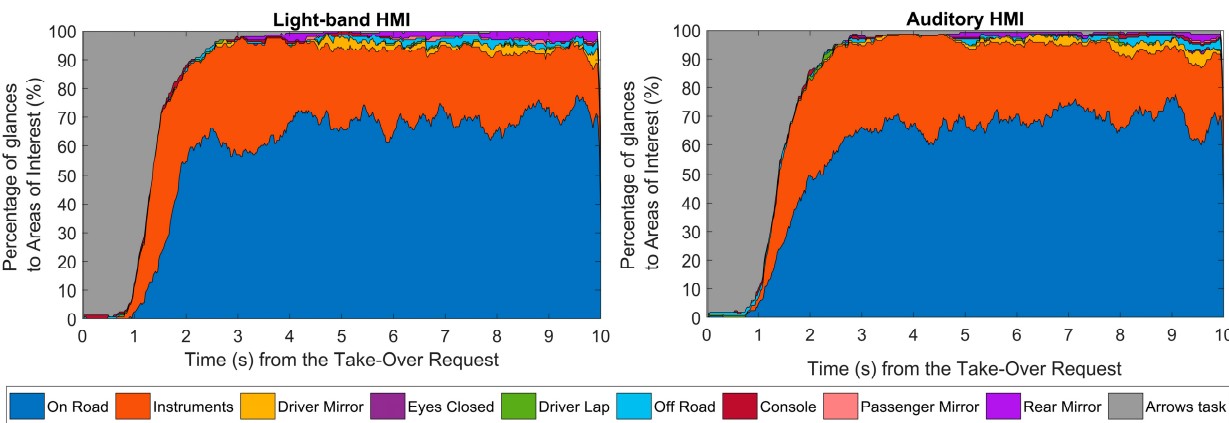

**Figure 6.** Drivers' visual attention allocation to different Areas of Interest in the 10 s after receiving a takeover request (TOR) for the Light-band HMI (**Left**) and Auditory HMI (**Right**) conditions.

An initial analysis of drivers' visual attention distribution to different AoIs showed that both groups took a similar time to shift their attention away from the Arrows task in response to the TORs, glancing primarily between the On Road area and Instrument Cluster. The only observable difference was a small increase in the proportion of glances to the Instrument Cluster (and an associated decrease in glances to the On Road area) approximately 1–2 s after the TOR for the Auditory HMI condition.

We also conducted a Markov-chain gaze transition analysis (see [28]) between areas of interest to investigate any differences between the two HMIs in terms of where drivers were most likely to look first after receiving a TOR. Results confirmed that, for both HMIs, the majority of drivers' first fixations after the Arrows task were directed towards the Instrument Cluster or On Road areas (Table 3). However, drivers were significantly ($p < 0.0001$) more likely to glance towards the Instrument Cluster first after receiving the Auditory TOR (65%), compared to the Light-band TOR (50%). Conversely, drivers were significantly ($p = 0.004$) more likely to glance On Road first after receiving the Light-band TOR, (35%), compared to the Auditory TOR (25%).

To measure the physical response time to the TORs, we assessed hands on wheel time and automation disengagement time. Hands on wheel time was taken from the onset of the TOR to the point at which both hands were detected on the steering wheel. Automation disengagement time was taken from the onset of the TOR to the point at which the driver disengaged the automation using the stalk, and the driving mode turned to manual. We conducted two repeated-measures ANOVAs (2 (HMI Type) × 5 (Takeover Number)), to investigate the effect of HMIs and takeover number on drivers' hands on wheel time and automation disengagement time, respectively.

As shown in Table 4 and Figure 7, there was no effect of HMI type on hands on wheel time, although the mean RT was higher for the Light-band HMI condition (M = 2.16 s, SD = 0.14), compared to the Auditory HMI condition (M = 1.93 s, SD = 0.07). There was no effect of takeover number, and no interactions.

**Table 3.** Markov-chain analysis of the location of drivers' first fixation away from the Arrows task after a TOR.

| | Console | Driver Lap | Driver Mirror | Instrument Cluster | Off Road | On Road | Passenger Mirror | Rear Mirror |
|---|---|---|---|---|---|---|---|---|
| Mann–Whitney U | 11,751 | 12,508 | 12,525 | 10,430 | 12,449 | 11,225 | 12,450 | 12,525 |
| Wilcoxon W | 23,076 | 26,536 | 23,850 | 24,458 | 26,477 | 22,550 | 23,775 | 23,850 |
| Z | −2.04 | −0.11 | 0.00 | −2.98 | −0.23 | −2.01 | −0.95 | 0.00 |
| p | 0.04 | 0.91 | 1.00 | <0.001 | 0.81 | 0.04 | 0.34 | 1.00 |
| Average Probability (Light-band HMI) | 10% | 0% | 0% | 50% | 5% | 35% | 0% | 0% |
| Average Probability (Auditory HMI) | 5% | 0% | 0% | 65% | 6% | 24% | 0% | 0% |

**Table 4.** Statistical results for takeover analysis.

| Effect | HMI Type | | | Event Number | | | HMI Type X Event Number | | |
|---|---|---|---|---|---|---|---|---|---|
| | F(*df*1, *df*2) | p | $\eta_p^2$ | F(*df*1, *df*2) | p | $\eta_p^2$ | F(*df*1, *df*2) | p | $\eta_p^2$ |
| Hands on wheel time | 3.443 (1,29) | 0.074 | 0.106 | 1.553 (4,29) | 0.19 | 0.051 | 1.265 (4,116) | 0.228 | 0.042 |
| Automation disengagement time | 0.364 (1,38) | 0.55 | 0.010 | 0.284 (4,38) | 0.88 | 0.007 | 0.302 (4,152) | 0.867 | 0.008 |

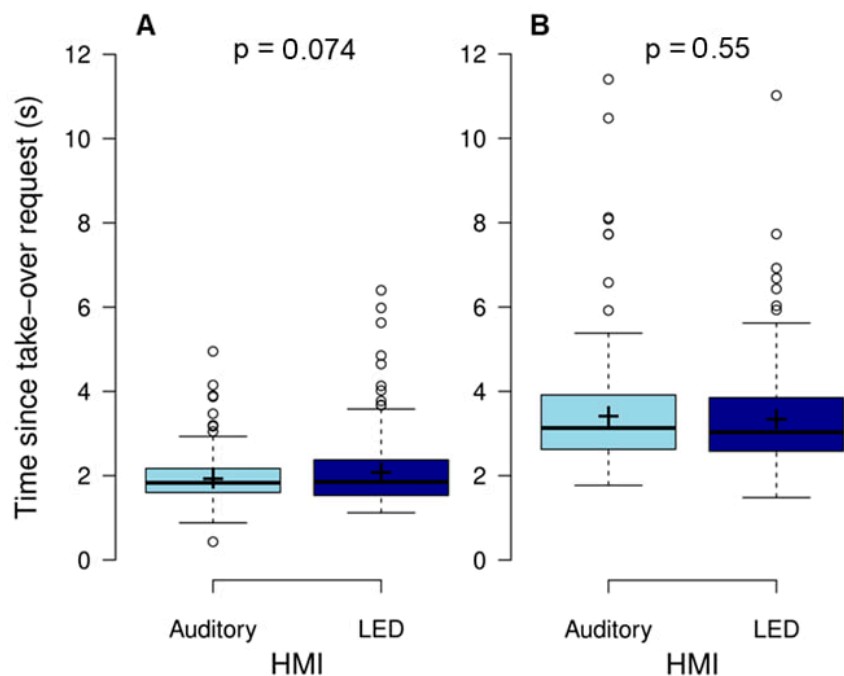

**Figure 7.** Time taken from TOR to (**A**) both hands on wheel and (**B**) automation disengagement for each HMI. Centre lines show the medians; box limits indicate the 25th and 75th percentiles as determined by R software; whiskers extend 1.

There was also no effect of HMI type on automation disengagement time, where the means for the Light-band HMI (M = 3.46 s, SD = 0.15) and Auditory HMI (M = 3.36 s, SD = 0.14) conditions were similar (Table 4, Figure 7). There was no effect of takeover number and no interactions.

*3.3. Perception of Safety and Trust*

Participants were asked to provide a perceived trust rating on a 5-point scale (Strongly disagree to Strongly agree) before the experiment and after each drive. The statement they were asked to rate was 'I trust/trusted that the vehicle will drive safely, while I do/did the Arrows task'. A one-way ANOVA was conducted to compare differences in the three perceived trust ratings for the baseline (i.e., pre-experiment) and post-drive ratings (with the Light-band and Auditory HMIs). There was no significant difference between the conditions (F(2,78) = 2.86, *p* = 0.084, $\eta p^2$ = 0.068; Figure 8).

Participants were also asked to rate 'I felt safe while doing the Arrows task during automated driving' on a 5-point scale (Strongly disagree to Strongly agree, Figure 8). We compared responses using a Kruskal–Wallis test. Overall, participants reported feeling safe while doing the Arrows task during automated driving, but there was no difference between the Light-band and Auditory HMI conditions [Z (1, 40) = 0.37, *p* = 0.713].

At the end of the second drive, we also asked drivers whether there was anything apart from takeover requests that interrupted their engagement in the Arrows task while

automation was on. 76% of participants (n = 31) reported that either a lead vehicle entering the lane or their vehicle braking interrupted their engagement in the Arrows task, confirming that the kinematic cue of the braking lead vehicle was felt by these drivers.

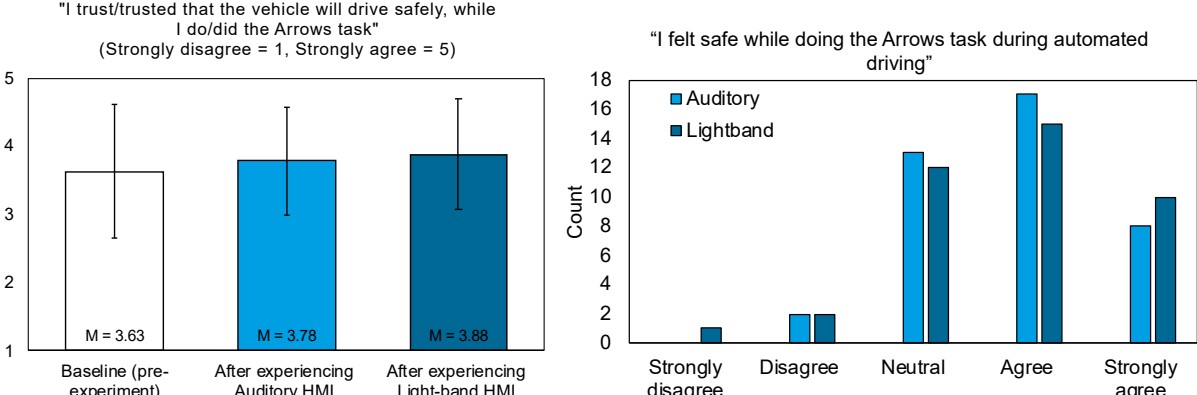

**Figure 8.** Mean scores of participants' responses to the statement "I trust/trusted that the vehicle will drive safely, while I do/did the Arrows task" (**Left**) and distribution of participants' responses to the statement "I felt safe while doing the Arrows task during automated driving" (**Right**). Note: The Auditory HMI was only used during takeover requests.

## 4. Discussion

Results showed that there were no major differences for the two HMIs in terms of the pattern of drivers' eye movements during the cut-in events, with a similar level of glances away from the NDRT and towards the forward roadway for both HMI conditions. This result suggests that the same degree of trust was bestowed to the automated system, regardless of the HMI condition. Regardless of HMI, the highest number of fixations to the road centre was seen during the third cut-in event (hard brake), which suggests that the kinematic cue was noticeable during this condition, with less reliance on the HMI. Overall, these results go against our initial hypothesis, which predicted that the presence of a peripheral Light-band, indicating that the automated system was engaged and fully functional, would put drivers more at ease, reducing the need to look up during the cut-in events to check automation status. Results from the subjective trust scales also suggest that the two HMIs were trusted equally.

In terms of the distribution of eye gaze at TOR, for both HMI conditions, visual attention was distributed between the road centre (On Road) and the Instrument Cluster areas (with the latter stating clearly whether drivers were in control of the vehicle). Therefore, for both HMI conditions, drivers initially prioritized glances to the Instrument Cluster over the On Road area, after the takeover alert, to confirm their role in the takeover. Similar behaviour has been found by [28–31], who suggested that drivers look at the dash area to receive information about system status, and their own role, before taking an action at TOR. However, our Markov-chain analysis showed that this behaviour was more pronounced when drivers only received an auditory takeover alert, possibly because the auditory alert provided no contextual information and, therefore, while it removed drivers' attention away from the Arrows task, it did not provide any further information about automation status. In contrast, the different colours and patterns of the Light-band HMI provided drivers with additional cues about the automation status, encouraging more drivers to glance On Road first, in favour of the Instrument Cluster. It must be noted that neither HMI concept (Light-band or Auditory) are a substitute for a standard cluster HMI, as, in both cases, drivers still relied on the instrument cluster to check the status of the automated system.

Previous studies investigating gaze patterns during transition of control [3] have found that drivers who were looking at the forward roadway during a TOR were less likely to crash into a lead vehicle, compared to those who were looking to other areas, including the

Instrument Cluster. Therefore, the use of supportive information from such peripherally presented Light-bands may be beneficial for directing drivers' attention to more safety-relevant areas during a TOR, such as the forward roadway (as also observed in [24]). However, further work needs to be done in this context to understand the value of these ambient messages for more critical takeovers, and for a wider range of scenarios, including those which require a response to other areas of the road and not just the forward roadway.

Regarding drivers' performance during the takeover request, there was no significant difference in time to resume control between the two HMI conditions. Further work is required in this context to establish the value of each HMI in assisting drivers for more critical TORs and scenarios, such as those that may lead to a crash if response is too slow. It is important to note that the delay seen in drivers' automation disengagement, which was after they placed their hands on the wheel, reflects the time taken to meet the three criteria of the driver disengagement algorithm. These results demonstrate that it is possible to elicit efficient takeovers that are also safe by using driver monitoring technologies to ensure drivers are following a series of actions while actively disengaging automation. However, it is equally important to support drivers who take a long time to resume control. In the current study, some drivers took longer than 10 s to disengage automation. Video analysis confirmed that, in these cases, drivers struggled to meet the three criteria of the driver disengagement algorithm. Therefore, more research is needed to understand the optimal parameters for such a disengagement algorithm to ensure that drivers are not unnecessarily hindered from resuming control. At the same time, this technique shows some value in the use of such algorithms, for avoiding unintended takeovers by drivers, while also ensuring that the driver is indeed ready and alert for a takeover.

## 5. Conclusions

The results of this study showed that driver behaviour and response to a Light-band takeover request was similar to that of an auditory alert. Drivers' level of engagement in the NDRT was also similar for the two takeover HMIs, and eye-tracking data showed that the same proportion of drivers looked at the road ahead during the harsh deceleration event (caused by the lead vehicle), regardless of HMI condition. Drivers' trust levels for the ADS were also quite similar for each drive. These results are somewhat in contrast to our original hypothesis, as we assumed that the presence of the ambient Light-band, which confirmed the automation status, would put drivers more at ease, increasing their trust in the system and precluding them from looking up at the driving scene when the L3 vehicle produced a harsh deceleration. Further work is warranted in this context to establish if more prolonged experience with such ambient lights increases drivers' trust and sustains their engagement in NDRTs. On the other hand, since the Light-band was at least as informative as the Auditory HMI, the use of this sort of stimulus in the vehicle may help reduce the use of the number of sound-based alerts and messages in vehicles.

It is acknowledged that simple light-band displays may not be suitable for conveying complex system-related information to drivers, which is in slight contrast to the conclusions of [24]. However, this type of ambient stimulus may be effective for attracting drivers' peripheral attention during NDRT engagement in L3 vehicles, after which more detailed information about system status and driver responsibility can be given by other displays, such as dash-based HMIs.

In terms of study limitations, the current study only used non-critical takeover scenarios after relatively short periods of automation. Therefore, future research should consider how response to these HMIs differs for more critical scenarios. Future research should also evaluate the value of such peripheral ambient displays for a range of messages and more strategic information, such as system-induced changes and events. Finally, this study investigated the value of a visual and auditory stimulus for takeovers in isolation. Future work should consider how the combination of these stimuli will affect driver takeover response and behaviour in such L3 driving conditions.

**Author Contributions:** Conceptualization, T.L., Y.M.L., R.M. and N.M.; Methodology, R.C.G., T.L. and N.M.; Software, J.K. and M.L.; Formal analysis, R.C.G.; Investigation, R.C.G. and T.L.; Resources, J.K. and M.L.; Writing—original draft, R.C.G. and T.L.; Writing—review & editing, R.C.G., Y.M.L. and N.M.; Supervision, Y.M.L. and N.M.; Project administration, N.M. All authors have read and agreed to the published version of the manuscript.

**Data Availability Statement:** Not applicable.

**Acknowledgments:** This research was developed in partnership with Toyota Motors Europe (TME). We would like to thank the institution for their financial support, as well as their input on the development of the experiment, that made this study possible.

**Conflicts of Interest:** The authors declare no conflict of interest.

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
