# Peer review of "Is Users’ Trust during Automated Driving Different When Using an Ambient Light HMI, Compared to an Auditory HMI?"

_information, doi:10.3390/info14050260_

Round 1

Reviewer 1 Report

This work details an HMI comparison for Automated Driving features for SAE L3. Overall, the work is well detailed, with a proper methodology and recruitment process. The experiment design is also adequate for the goals that the authors wan to achieve.

The work is considered to be appropriate for publication, although several minor issues should be addressed:

-There are missing data (keyworks).

-Cross references in pages4, and 5

-Mercedes Bens is Mercedes Benz

-the last paragraph in line 376 seems to have been misplaced.

As stated in the  conclusions, the comparison data seems to indicate that Auditory and Visual HMI concepts are similar in effectiveness. A very interesting approach would have been to compare these individual approaches with an integrated one (vistual+ auditory), and see if the performance is significantly changed.

Author Response

Please. see the attachemnt

Reviewer 2 Report

The goal of this study was to assess whether an ambient peripheral light display could be 10 used to improve drivers' perceived safety and trust during L3 automated driving. The authors conducted a simulator study with 41 participants to compare an ambient light display to an auditory display. By comparing participant’s trust, takeover performance, and gaze behavior, the authors have concluded that there were no differences between the two displays, except the ambient light display produced more glances towards road center.

I believe this paper can add value to the literature. My specific comments in order of presentation are below.

1.     There are no keywords.

2.     The authors compare an ambient light display to “a more traditionally used auditory alert” – line 62, pg. 2. However, are the two really comparable? The ambient light display should be compared to another visual display. Also, multimodal displays, e.g., an ambient light display plus an auditory alert, are more common and suggested by the literature. (On pg. 5, lines 203-209, the reader discovers that in fact a visual warning was displayed at the same time as the auditory or ambient light display.) The authors should justify why they decided to compare a single modality, dual visual display (ambient light + dashboard visual warning) to a multimodal display (auditory + dashboard visual warning).

3.     There is an incorrect cross-referencing link on line 155, pg. 4. This is a recurring issue throughout the paper.

4.     While Figure 1 is helpful, a more accurate depiction of the interior of the vehicle would be more useful to the reader.

5.     Though a “score to beat” was displayed to participant’s to keep them engaged (line 192, pg. 5), were participants incentivized to complete the task? It seems possible that a participant could never engage in the task (since there were no consequences) and therefore, never be sufficiently distracted.

6.     The authors should explain their trust question (in table 2). Was it a reliable and valid measure? Why not use a standard measure of trust, e.g., by Jian et. al, 2000?

7.     While figure 6 is interesting, it is not connected to the text. Does it relate to lines 364-366, pg. 11?

8.     Lines 376-378, pg. 11 do not belong.

9.     Line 405, pg. 14: what is Oerceotuibs?

10.  Overall, the statistical results and analysis seem reasonable, but given the sample size of 41, there are many statistical tests. I encourage the authors to consider the effect of inflated Type I error, or consult a statistician, when conducting so many statistical tests.

11.  Though this reviewer wants to support the publication of a null finding, which is often not reported, more text should be dedicated towards the implications of these findings. I suggest that the authors add a Conclusion or Implications section at the very end of the article.

Round 2

Reviewer 2 Report

Thank you for addressing my comments.

There is still an incorrect cross referencing link on pg. 6, line 194.

Author Response

Thank you for your review. The correction was made in Line 194 (highlighted in yellow for easier tracking).